# Mangrove Forest Cover Change in the Conterminous United States from 1980–2020

**Chandra Giri** [1,*] **, Jordan Long** [2] **and Prapti Poudel** [1]

1   Office of Research and Development, United States Environmental Protection Agency, 109 T.W. Alexander Drive, Research Triangle Park, Durham, NC 27709, USA; giri.prapti@epa.gov

2   Earth Resources Observation and Science (EROS) Center, 252nd Street, Sioux Falls, SD 47914, USA; jlong@usgs.gov

*   Correspondence: giri.chandra@epa.gov; Tel.: +1-919-224-9817

**Abstract:** Mangrove forests in developed and developing countries are experiencing substantial transformations driven by natural and anthropogenic factors. This study focuses on the conterminous United States, including Florida, Texas, and Louisiana, where coastal development, urbanization, hydrological pattern alterations, global warming, sea level rise, and natural disasters such as hurricanes contribute to mangrove forest changes. Using time-series Landsat data and image-processing techniques in a cloud computing platform, we analyzed the dynamics of mangrove forests every five years from 1980 to 2020. Each thematic product was independently derived using a region of interest (ROI) suitable for local conditions. The analysis was performed using consistent data sources and a unified classification methodology. Our results revealed that the total mangrove area in the conterminous United States (CONUS) in 2020 was 266,179 ha. with 98.0% of the mangrove area in Florida, 0.6% in Louisiana, and 1.4% in Texas. Approximately 85% of the CONUS mangrove area was found between 24.5° and 26.0° latitude. Overall, mangrove forests in the CONUS increased by 13.5% from 1980 to 2020. However, the quinquennial variation in aerial coverage fluctuated substantially. The validation of 2020 using a statistical sample of reference data confirmed the high accuracy of 95%. Our results can aid policymakers and conservationists in developing targeted strategies for preserving the ecological and socio-economic value of mangrove forests in the conterminous United States. Additionally, all the datasets generated from this study have been released to the public.

**Keywords:** mangrove forests; remote sensing; image processing; Landsat; change analysis





## 1. Introduction

Mangrove forests in both developed and developing countries are undergoing significant transformations due to both natural and anthropogenic factors. In the conterminous United States encompassing Florida, Texas, and Louisiana, the forests are changing due to coastal development, urbanization, alterations in hydrological patterns, the effects of global warming, sea level rise, and the occurrence of natural disasters such as hurricanes [1–3].

The mangrove forests in various parts of the world contribute to ecosystem services to varying degrees, depending on the specific economic and ecological conditions present in each location [4,5]. In the continental United States, mangrove forests provide invaluable ecosystem goods and services that extend beyond their ecological significance, offering benefits in terms of coastal protection, biodiversity conservation, carbon sequestration, water filtration, fisheries support, and recreational opportunities [6–8].

In the face of diminishing forest resources and ecosystem goods and services, the remaining mangrove forests and their invaluable services are increasingly vulnerable to the compounding pressures of economic and biophysical factors [9–11]. Effective quantification of the rates, patterns, causes, and consequences of mangrove forest cover changes heavily relies on using time-series remote-sensing data. Such data play a crucial role in

understanding and monitoring the dynamics of mangrove forests and their response to various drivers of change.

In this context, our research aimed to conduct a spatiotemporal analysis using Landsat satellite data spanning five-year intervals from 1980 to 2020. The goal was to assess both positive (afforestation) and negative (deforestation) changes in mangrove forest coverage within the contiguous United States.

## 2. Study Area, Data Basis, and Methods

### 2.1. Study Area

Mangrove forests of the conterminous United States (CONUS) are exclusively situated in the Gulf of Mexico states of Texas, Louisiana, and Florida (Figure 1). South Florida is home to the largest tract of continuous mangrove forests in the United States, of which 75% of the country's mangroves occur within the Everglades National Park alone [3]. These forests are predominantly composed of three mangrove species: red mangrove (*Rhizophora mangle*), black mangrove (*Avicennia germinans*), and white mangrove (*Laguncularia racemosa*). The mangrove forests in Texas and Louisiana mainly consist of black mangroves.

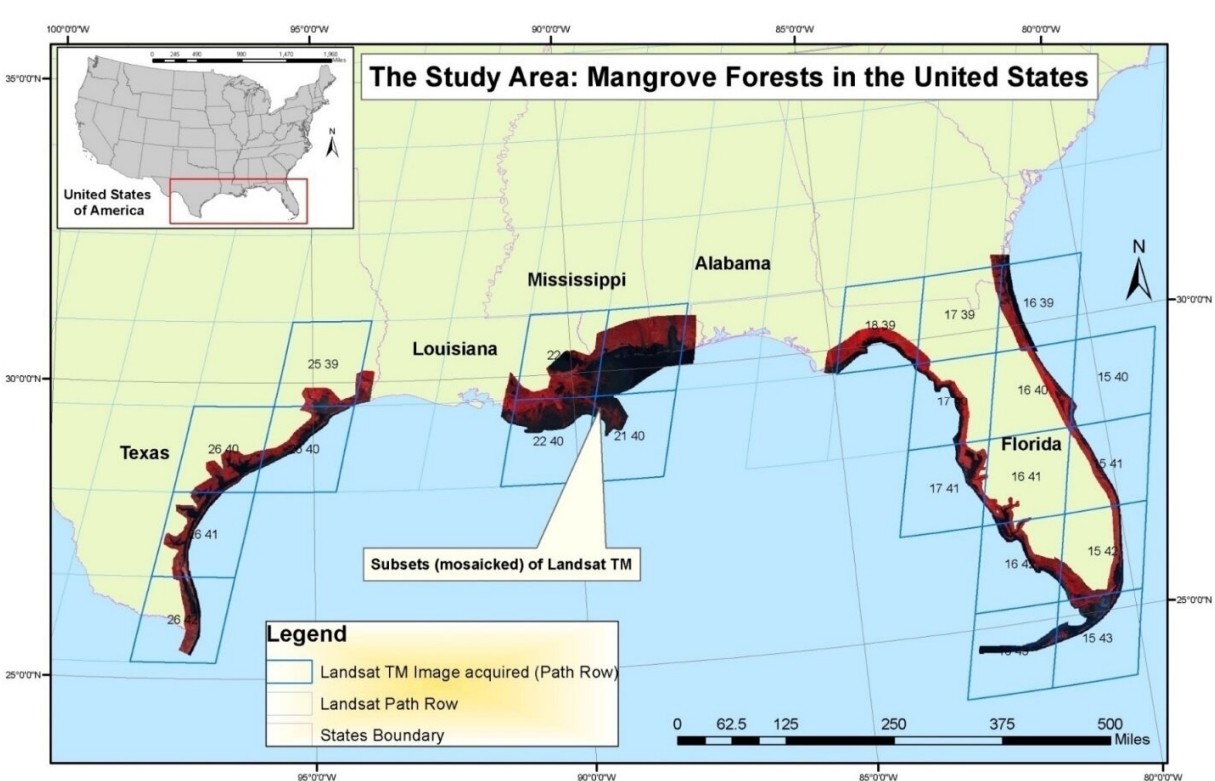

**Figure 1.** Approximate location of mangrove habitats in CONUS.

Mangrove forests in the United States are believed to be expanding towards the pole due to climate change [12]. Warmer winter air temperatures and less extreme sub-freezing winter air temperatures are likely causes of the expansion of black mangroves (*Avicennia germinans*) into salt marshes dominated by smooth cordgrass (*Spartina alterniflora*) [8,13,14]. However, long-term data dating back to the 1980s suggests the northernmost latitudinal limit of mangrove forests along the east and west coasts of Florida, in addition to Louisiana and Texas, has not systematically expanded toward the pole [15]. The forest is, however, expanding within the geographic limit [15]. Other factors such as fire, tropical storms, sea level rise, sedimentation/erosion, changing hydrology, and specific management practices are also believed to be contributing to the expansion in localized areas [16,17].

### 2.2. Data and Methods

The Landsat Multi Spectral Scanner (MSS), Thematic Mapper (TM), Enhanced Thematic Mapper Plus (ETM+), and Operational Land Imager (OLI) Tier 1 top-of-atmosphere (TOA) reflectance data were procured from the collections of the United States Geological Survey (USGS) and Google Earth Engine (GEE) archive. The collaborative Landsat program, initiated in the early 1970s by both the National Aeronautics and Space Administration (NASA) and USGS, represents the world's most extensive and uninterrupted repository of satellite-based Earth observation data. The complete data archive is openly accessible through USGS and is also made available on various other platforms including the Google Earth Engine. This study utilized data from the Landsat series corresponding to the years circa 1980, 1985, 1990, 1995, 2000, 2005, 2010, 2015, and 2020.

The Tier 1 Landsat data used in this study have undergone Level-1 Precision Terrain (L1TP) processing, which ensures well-defined radiometry and consistent inter-calibration across various Landsat sensors [18]. Additionally, Tier 1 scenes are meticulously georegistered and maintained in conformity with specified tolerances, ensuring a root mean square error (RMSE) of 12 meters or less. The specifications for the reflectance product and the constants applicable to all Landsat sensors including TM, ETM+, and OLI were comprehensively detailed and tabulated by Chander et al. (2009) [19].

It is important to note that all Tier 1 Landsat data are considered suitable for land-cover mapping and monitoring [19]. These data can be used to detect and estimate long-term dynamic land-cover change dating back to the 1970s [18,19].

We employed a cloud compositing technique to create mosaic images devoid of clouds for every five-year period from 1980 to 2020. This method utilizes a maximal value approach for each spectral band [20,21]. In situations where cloud-free data were not accessible for these specific five-year periods, Landsat data from one or two years prior to or after the target date were utilized. The increased availability of Landsat data over the years provided an opportunity to select data from the target date itself.

For the 1980 mosaic, we utilized Landsat-3 MSS images, which possess a spatial resolution of 60 m. To ensure compatibility with other timeframes, these images were resampled from 60 m to 30 m using the cubic convolution resampling technique. It is important to note that this resampling does not enhance the original spatial resolution of Landsat MSS data. In contrast, all other Landsat datasets consisted of TM or OLI images, boasting a spatial resolution of 30 meters. Since the launch of Landsat 4 in 1982, satellite data have been consistently captured at a uniform spatial resolution of 30 m per pixel, along with similar spectral bands. This uniformity facilitates a comprehensive multi-decadal analysis of both land cover and land use.

Training points were selected manually in the Landsat composite for each epoch by dividing the scene into ROI. ROI were selected based on the likelihood of mangrove presence or absence based on historical data. This method of ROI selection has been demonstrated to simplify image classification algorithms by reducing variability within the image [1,3].

The distinctive signature of mangrove forests within satellite data facilitated the identification of training points. The number of training points within each ROI differed, and this was an interactive procedure. Training samples were subject to modifications (additions or removals) based on a visual inspection of the classification outcomes. This adjustment process continued until a satisfactory classification outcome was achieved. In addition to the Landsat composite, previously published mangrove maps and very high-resolution satellite data of the area were utilized as points of reference.

Three major classes were selected for the classification: mangrove, non-mangrove, and water bodies. Brief descriptions of these classes are described in Table 1.

**Table 1.** Classification criteria.

| Land Cover Types | Description |
|---|---|
| Mangrove | True mangroves that grow in brackish and saline water. In the United States, three main species are found: red, black, and white mangroves |
| Non-mangrove | All land use/land cover classes other than mangrove and water bodies including cropland, urban areas, barren land, shrubland, and grassland |
| Water bodies | Ocean water, brackish water, perennial river, streams and water reservoirs, and open water like lakes and ponds. |

The classification was performed within each ROI due to unsatisfactory classification outcomes when attempting to classify expansive regions encompassing the entire image. As mentioned earlier, by employing ROIs, the heterogeneity within the Landsat image was mitigated, leading to improved classification results.

We used the JavaScript version of the code available from the GEE platform (https://code.earthengine.google.com/ for image classification, accessed on 16 June 2023). The GEE platform inherits several supervised classification algorithms that can be employed for image classification. The GEE and other cloud computing platforms are becoming standard for land-cover mapping and monitoring [22,23]. We used the pixel-based random forest (RF) classification algorithm. The RF is a tree-based classifier that includes K-decision trees [24]. It is a combination of multiple decision trees and is widely referred to as an ensemble learning technique. This classifier is a well-known and powerful classification model used to model complex non-linear processes with a lesser computational burden. The algorithm overcomes the problem of overfitting by constructing an ensemble of decision trees. Numerous studies have highlighted the accuracy and superior performance of RF classifiers in the realm of land-cover classification [23–26].

During image classification, all bands except the thermal infrared band were used. The thermal band was not used primarily due to the coarse spatial resolution (120 m).

After classification, all of the ROIs were mosaicked for Texas, Louisiana, and Florida in ERDAS Imagine, where these three were mosaicked to produce seamless coverage for the conterminous United States.

Mangrove forest cover change statistics were generated in each 0.5° latitudinal range on the east and west coasts of Florida except for the southernmost area where only one 0.5° latitudinal range was present. Texas and Louisiana were also delineated by these 0.5° latitudinal ranges and change statistics were calculated. (Figure 2). The goal of this analysis was to identify the distribution and dynamics of mangrove forests in each latitudinal zone.

Post-classification change detection was performed for 1980–2020, as well at every five-year interval from 1980 to 2020. An overlay method was used to compute from-to change detection. In this process, independently generated classifications at two time intervals are compared. The main advantage is to limit issues associated with radiometric calibration (i.e., solar illumination, atmospheric absorption and scattering, and sensor properties) in between intervals. However, the accuracy of the initial classification of each era dictated the accuracy of the post-classification comparison. In effect, the end accuracy approximates the compounding of the accuracies of every single classification.

Yet another benefit of post-classification change detection is its ability to reveal the characteristics and extent of alterations in land cover that have occurred over a specific period. The from-to-change analysis proved valuable in offering in-depth insights into the specific types of land-cover changes. In contrast, other researchers' have used spectral change analysis techniques [27–29].

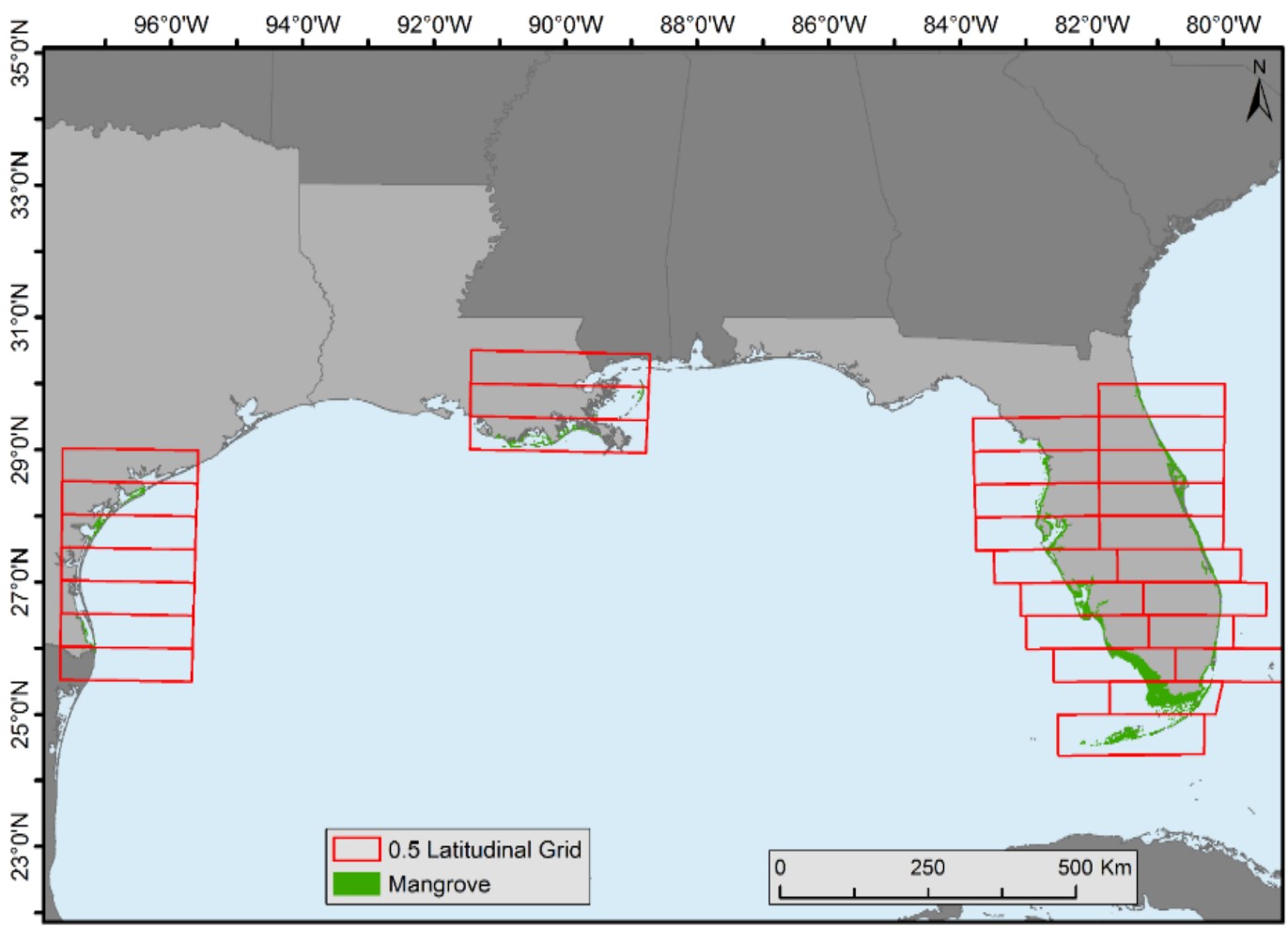

**Figure 2.** The 0.5° latitudinal zones in Florida, Louisiana, and Texas created for mangrove forest distribution and change analysis.

We generated a matrix reflecting shifts between different land cover classes. This was carried out at five-year intervals spanning from 1980 to 2020. The selection of a five-year time frame was driven by several factors: (i) the limited availability of cloud-free, high-quality images in the region, particularly during the early years of the Landsat archive, and (ii) the need to establish a consistent time series for the entire geographic area.

Specifically for the mangrove forests class, a comprehensive examination was conducted to ascertain the spatial extent and variations in area, including gains, losses, persisting regions, and transitions between different land-cover classes. Through cross-tabulation, an understanding of the fundamental change processes between mangrove and non-mangrove areas, as well as water bodies, was achieved.

We performed the validation in three steps: (i) comparison of our results visually with existing mangrove and land-cover maps, (ii) qualitative assessment, and (iii) quantitative validation. Published maps such as the National Oceanic and Atmospheric Administration (NOAA)'s Coastal Change Analysis Program (CCAP) and National Land Cover Dataset of the USGS were used for the first step. For the qualitative assessment, all of the classification maps were divided into a regularly spaced grid and checked visually with the goal to minimize gross errors during the image interpretation. For the quantitative validation of 2020, 100 stratified randomly sampled points were generated in ERDAS Imagine with a minimum of 25 points for each class. These points were compared with aerial photographs available in Microsoft Bing as reference data. Accuracy assessment measures (i.e., overall accuracy, producers' accuracy and users' accuracy) were computed from the classification-reference comparison.

## 3. Results and Discussion

We created a comprehensive database documenting the distribution and changes in mangrove coverage across the CONUS every five years spanning from 1980 to 2020. This endeavor involved utilizing uniform data sources, a standardized classification system, and a consistent methodology to carry out the analysis.

### 3.1. Mangrove Distribution

Figure 3 illustrates the geographical arrangement of mangrove forests within the contiguous United States (CONUS) during the year 2020. The overall expanse of mangrove coverage in the CONUS for 2020 encompassed 266,179 hectares, with the distribution being 98.0% in Florida, 0.6% in Louisiana, and 1.4% in Texas.

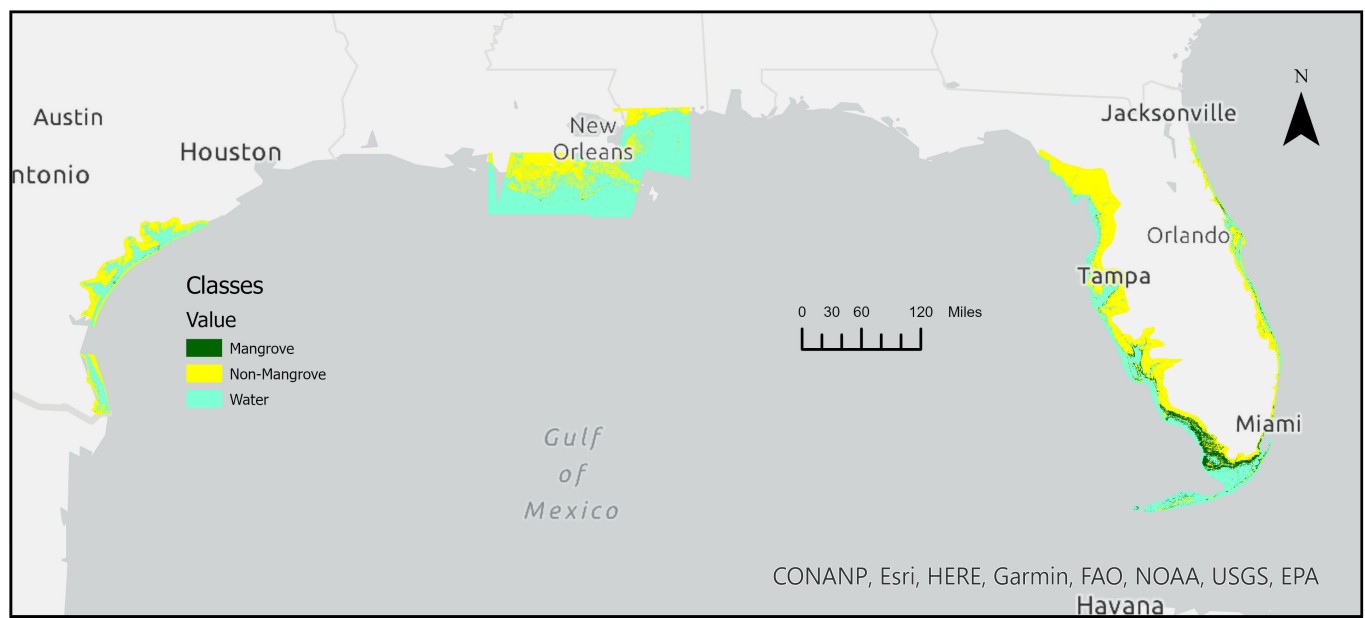

**Figure 3.** Spatial distribution of mangrove forests of the continental United States in 2020.

We also examined the distribution of mangrove forests along latitudinal gradients within the CONUS. In this analysis, statistics pertaining to changes in mangrove forest coverage were computed for each 0.5° latitudinal interval along the eastern and western coasts of Florida. However, it is worth noting that separation on the eastern coast was not feasible in the southernmost regions. Similar analyses were conducted for Texas and Louisiana, following the methodology outlined by Giri and Long (2016) [17]. Consistent with expectations, there was a general decline in the occurrence of mangroves as latitude increased from south to north. Notably, 85% of the total mangrove area in the CONUS was concentrated within the latitude range of 24.5° to 26.0°.

Alterations in the arrangement and prevalence of mangrove species, whether inside or beyond their traditional geographical boundaries, can yield significant repercussions for the array of ecological benefits and services they offer. In the CONUS, there is a prevailing notion that mangroves are advancing towards higher latitudes (northward) due to a reduction in the frequency and intensity of extreme cold events in recent years. Concurrently, the rise in sea levels frequently emerges as a contributing factor to the inland progression of mangroves on a local scale. Extensive documentation supports the idea that elevated temperatures expedite the expansion of mangroves and the accumulation of surface elevation in subtropical wetlands [30,31].

Employing four decades of satellite imagery and on-site observations focused on the CONUS, our findings indicate the following:

(i) The notion of a northward expansion of mangrove forests remains uncertain;

(ii) Simultaneously, a progressive expansion towards inland areas, within the established historical northern boundary, is presently underway.

Our findings illustrate that the northernmost latitudinal boundaries of mangroves along the east coast of Florida (81.317299°W, 29.945414°N) and the west coast of Florida (83.046396°W, 29.162046°N), as well as in Louisiana (88.860357°W, 30.038007°N) and Texas (96.410255°W, 28.428913°N), have not exhibited a consistent poleward expansion between 1980 and 2020 (as indicated in Table 2). Nevertheless, it is worth noting that the historical northern limit prior to the 1980s lacks comprehensive surveying or documentation, and the mapping of individual mangrove recruits was hindered by the lack of 30-meter resolution of Landsat satellite data.

**Table 2.** Geographic coordinates of the northernmost extent of mangrove distribution in eastern Florida, western Florida, Louisiana, and Texas.

| State | Latitude (in Decimal Degrees) | | Longitude (in Decimal Degrees) | |
|---|---|---|---|---|
| | 1980 | 2020 | 1980 | 2020 |
| Eastern Florida | 29.86373 | 29.94541 | −81.30328 | −81.31730 |
| Western Florida | 29.16205 | 29.16232 | −83.04640 | −83.046480 |
| Louisiana | 30.03801 | 29.97985 | −88.86036 | −88.83519 |
| Texas | 28.42891 | 28.43685 | −96.41026 | −96.40120 |

### 3.2. Mangrove Change

In the aggregate, mangrove forests in the CONUS exhibited a notable growth of 13.5% spanning from 1980 to 2020. The rise in mangrove coverage between 1980 and 2015 accounted for 4.3%, implying that the alterations from 2015 to 2020 surpassed those of the preceding 35 years. However, there were substantial oscillations in quinquennial (i.e., "every five years") aerial expansion, with the lowest mangrove coverage recorded in 1990 (172,300 hectares) following a severe winter sub-freeze incident in late 1989, while the zenith was observed in 2020 (266,179 hectares) (depicted in Figure 4). The impact of winter freezes on mangroves can be profound. For example, during the winter freeze of 2017/2018, Louisiana experienced degradation in approximately 90% of its mangrove population [30].

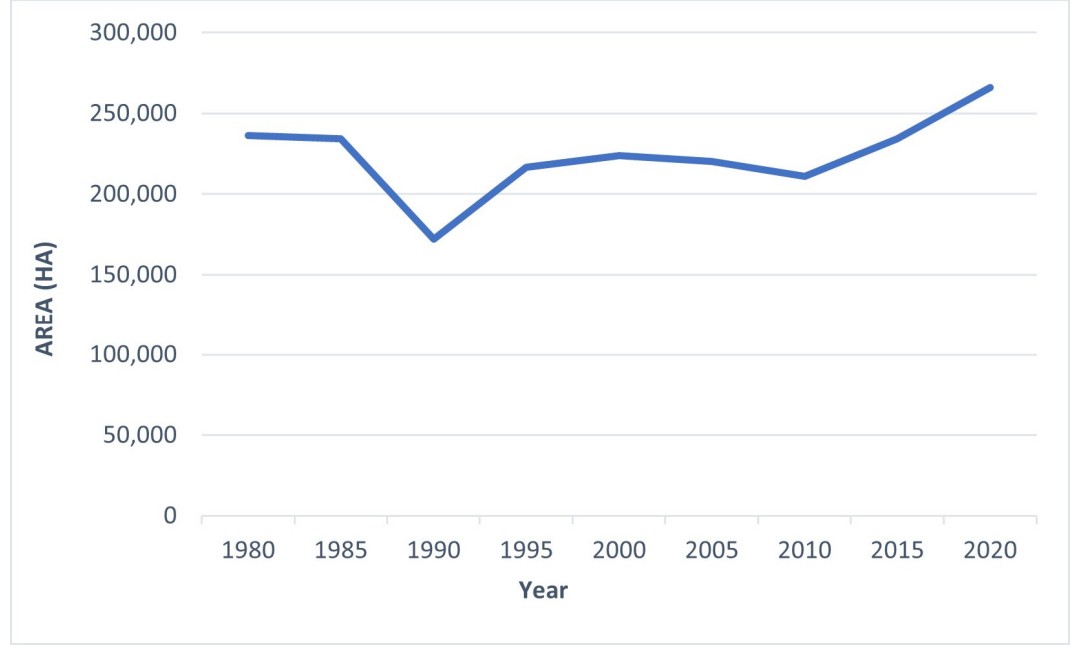

**Figure 4.** Mangrove forest areas every five years from 1980 to 2020 in the CONUS.

Figure 5 depicts the acreage of mangrove forest regions that remained unaltered, underwent deforestation, and experienced afforestation at five-year intervals spanning from 1980 to 2020. During each of these periods, more than 85% of the mangrove areas displayed constancy, except for the time frame between 1985 and 1990, where only 66% of the forests remained unchanged. Notably, the alteration patterns of both deforestation and afforestation areas varied across each era. Deforestation is characterized by the shift from mangrove to non-mangrove or from mangrove to water, while afforestation pertains to the transformation from non-mangrove to mangrove or from water to mangrove.

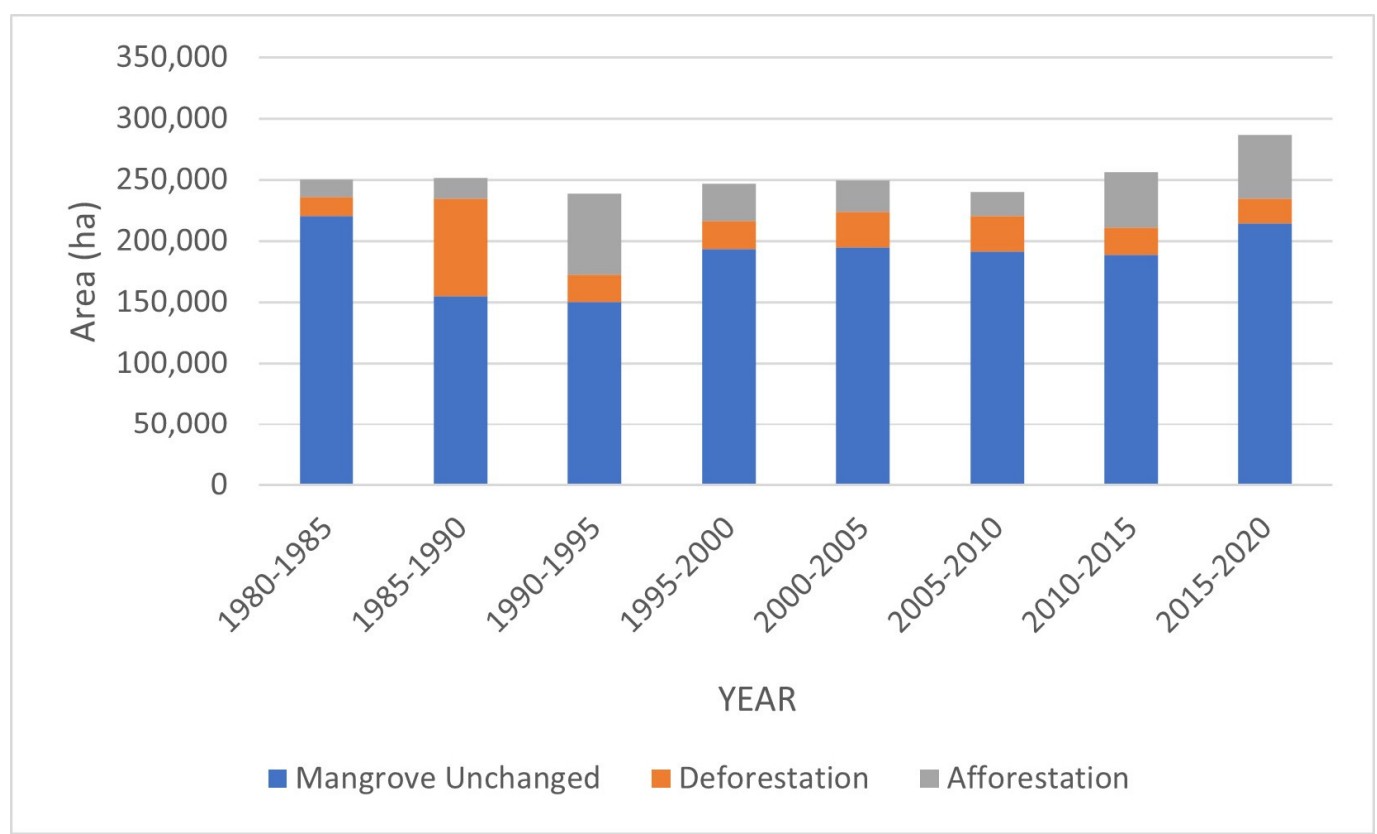

**Figure 5.** Comparison of mangrove areas unchanged, deforestation, and deforestation every five years from 1980–2020.

The observed changes, encompassing both deforestation and afforestation, exhibited irregularity along the latitudinal grid. Both mangrove expansion and reduction transpired across all latitudinal segments. Nevertheless, the alterations were more conspicuous in certain regions compared to others, as depicted in Figure 6.

### 3.3. Proximate Causes of Mangrove Change

Various factors, including but not limited to winter freeze, storms, land-use practices, hydrology, erosion, impoundment, sea level rise, and human plantation efforts, frequently contribute to alterations in mangrove forests (as illustrated in Figure 7). The influence of these factors is contingent upon the geographical areas in question (15), often with multiple factors concurrently shaping the observed changes.

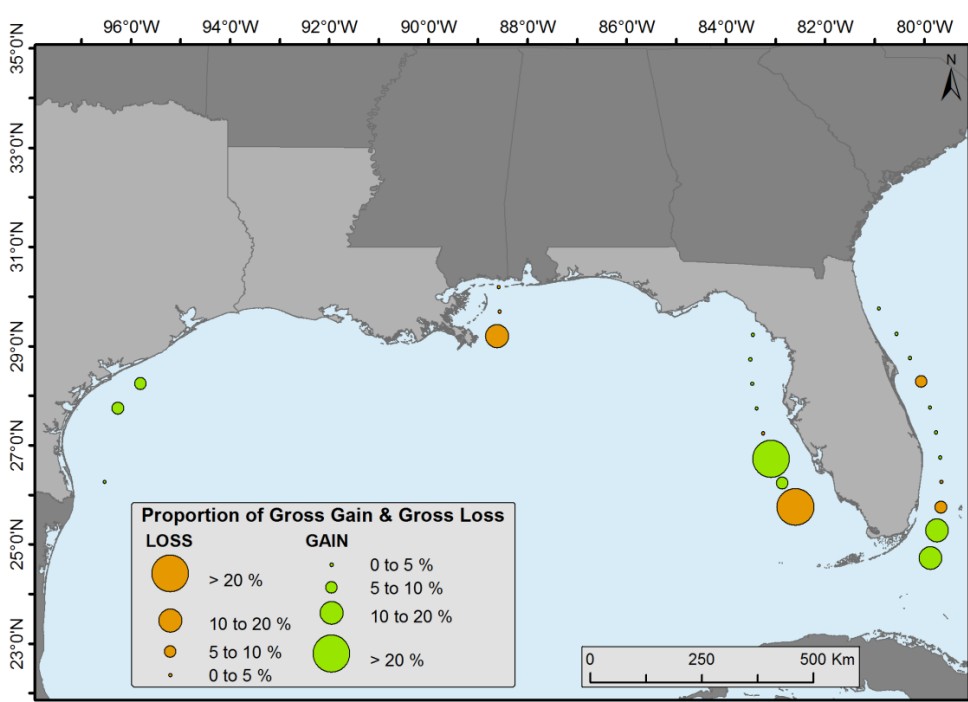

**Figure 6.** Proportion of total mangrove forest cover change per 0.5° latitudinal intervals during 1980 to 2020.

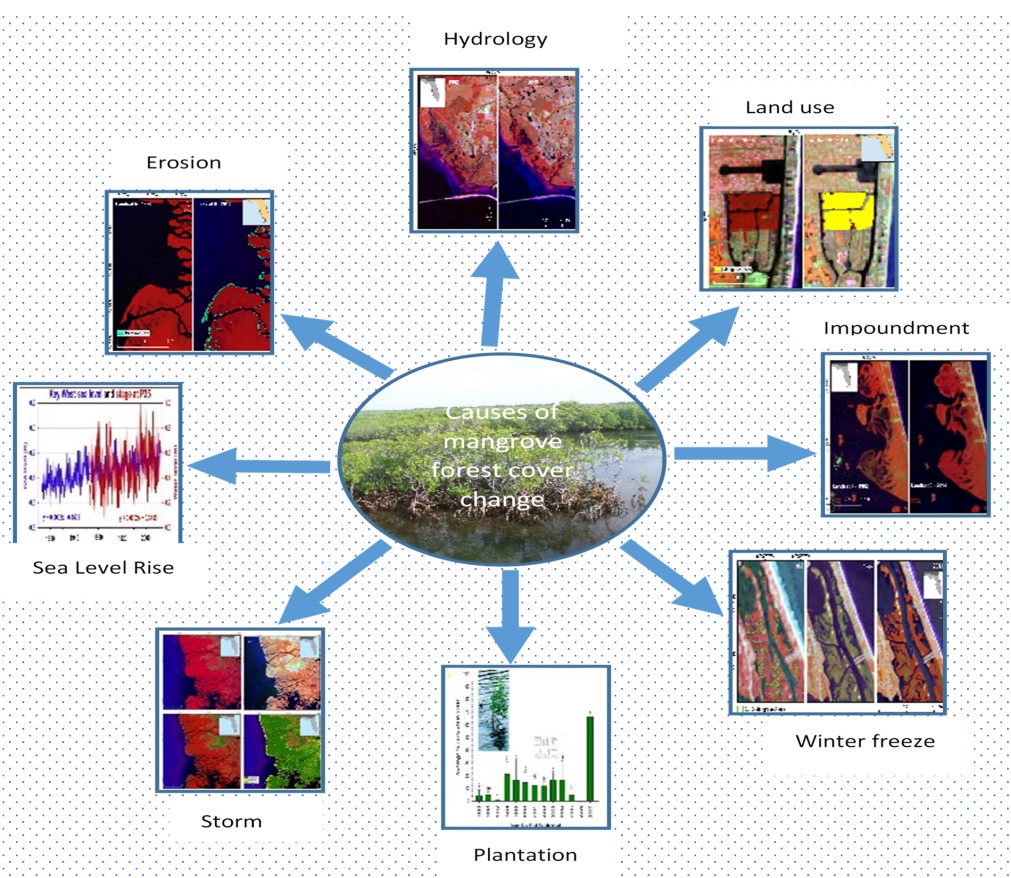

**Figure 7.** Proximate causes of mangrove forest cover change in the CONUS.

The role of winter freeze is pivotal in influencing the dynamics between mangrove and salt marsh ecosystems, particularly at the northern boundary of mangrove distribution

in the continental United States. This phenomenon is notably prominent along the eastern coast of Florida and Louisiana, and to some extent, also on the western coast of Florida and Texas. Instances of significant winter freezes in 1983, 1989, and 2010 have led to the demise of mangrove forests, particularly those dominated by black mangroves (*Avicennia germinans* (L.)) in much of the northern boundaries. In these areas, the mangrove presence has been supplanted by salt marshes, primarily characterized by the dominance of smooth cordgrass (*Spartina alterniflora* Loisel.). However, following a period of mild winters spanning a decade or more, mangroves have once again reclaimed these regions. This intricate interaction between mangroves and Spartina grasses has conceivably been unfolding within the CONUS over an extended timeframe. Our findings, however, have specifically documented this phenomenon after the year 1980.

The impact caused by the significant winter freezes in 1983, 1989, and 2010 exhibited variability in terms of damage. Nonetheless, the affected mangrove areas managed to recuperate, surpassing 90% of their initial levels within a span of 5 to 10 years. Figure 8 illustrates this finding after the substantial damage caused during the winter freeze of 1983 in Cedar Keys, Florida.

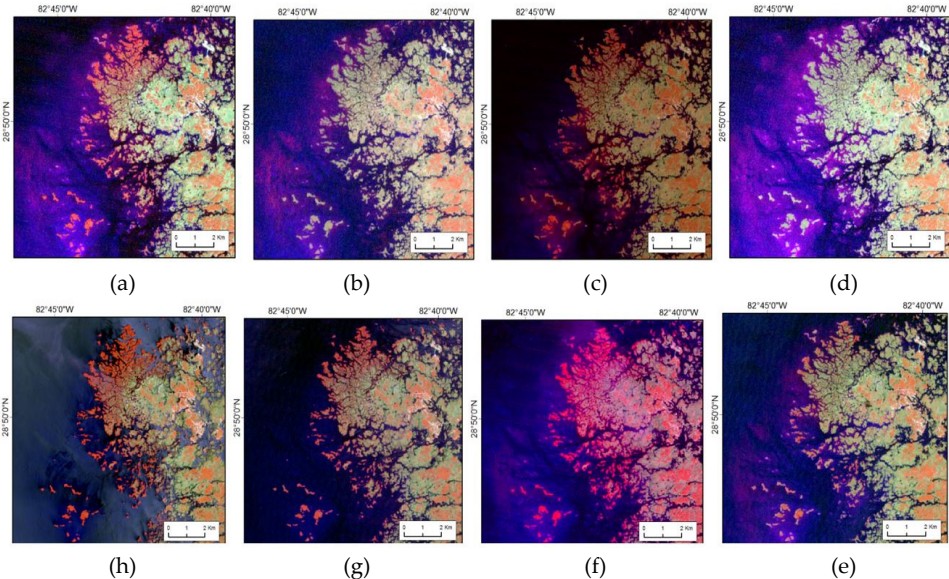

**Figure 8.** Mangrove damage due to severe winter freeze and subsequent recovery in Cedar Keys, Florida. The image is displaced in Landsat False Color Composite; dark red is mangrove forests, blue is water bodies, and other colors represent non-mangrove and non-water. (**a**) Landsat 1982, (**b**) Landsat 1984, (**c**) Landsat 1988, (**d**) Landsat 1990, (**e**) Landsat 1995, (**f**) Landsat 2008, (**g**) Landsat 2010, (**h**) Landsat 2014.

Mangrove forests within the CONUS are also frequently subjected to the impact of tropical storms and hurricanes. An illustrative instance is Hurricane Wilma in 2005, which inflicted significant harm upon a substantial portion of the mangrove forests along the Shark River within the Everglades National Park (Figure 9). The effects of tropical cyclones on mangrove forests span a spectrum from transient defoliation to extensive tree mortality. The passage of the hurricane of 2005 led to diverse outcomes: some trees lost their canopies, others endured breakage, uprooting, leaf detachment, and trunk fractures. However, despite the damage, mangrove regeneration managed to restore much of the impacted areas by 2010, except for certain coastal fringes, as indicated in Figure 9.

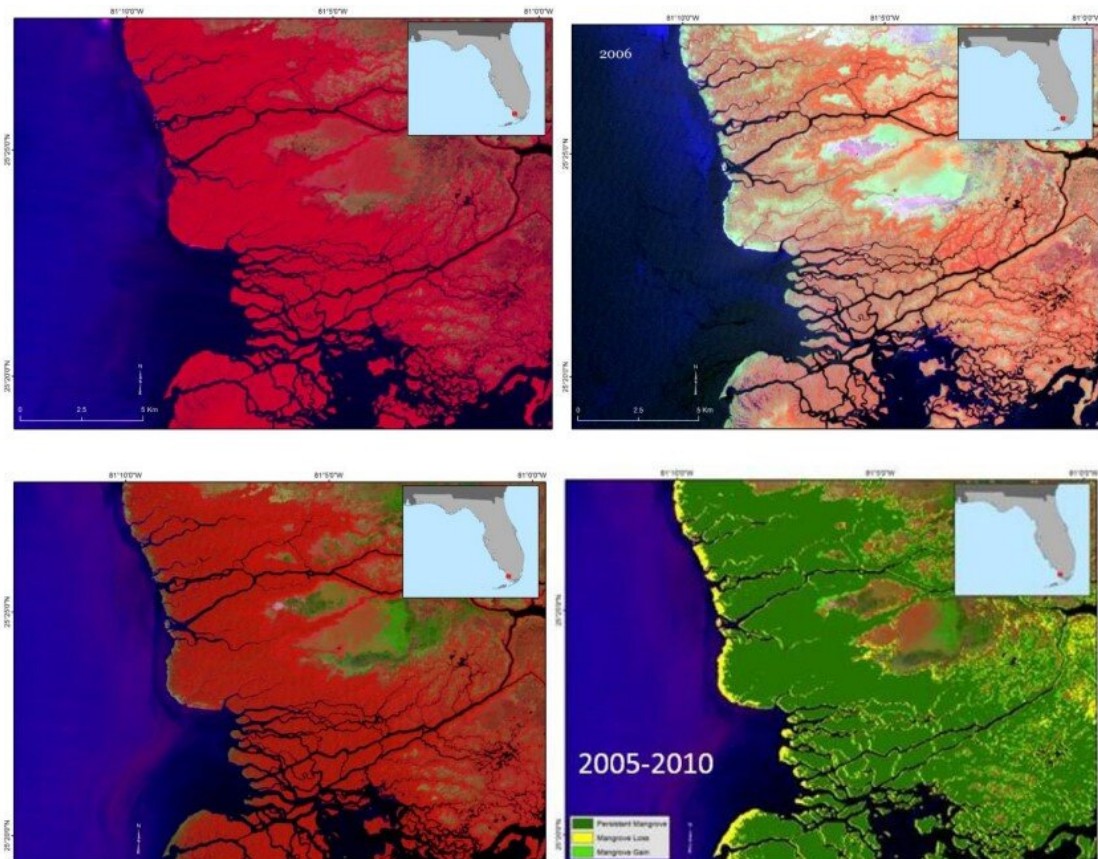

**Figure 9.** Mangrove damage just before hurricane Wilma in the winter of 2005 (**upper left**), extensive damage after Hurricane Wilma (**upper right**), and recovery in 2010 (**lower left**) as seen from the Landsat images. Landsat images are shown in False Color Composite, and dark red roughly corresponds to healthy mangrove vegetation. Much of the mangrove forest recovered except along the coastal fringes as seen in the change map from 2005 to 2010 (**lower right**).

Mangrove forests in the CONUS have also been impacted by other hurricanes, including Hurricane Andrew in 1992, Wilma in 2005, and Irma in 2017. However, the effects of the 1992 and 2017 hurricanes were not discernible in our study because much of the affected forest had already regenerated by the year of our observations, which were 1995 and 2020, respectively. This observation underscores the importance of conducting annual analyses of mangrove cover changes to effectively capture such dynamic shifts. Additionally, the damage caused by the 2005 hurricane is not evident in our study because it occurred later in the year and was not captured in the Landsat mosaic of 2005. These findings further highlight the potential need for future monthly monitoring of mangrove forests to comprehensively track their changes.

Other previous investigations have revealed that the subsequent recuperation from storm-induced damage varies, influenced by both the intensity of the storm and the underlying soil conditions [27–29]. Notably, certain storms deposit sediment rich in phosphorus, which in turn stimulates the growth of mangroves [29]. For instance, in the aftermath of Hurricane Irma, approximately 83% of the damaged mangrove forests exhibited recovery within the first year [30]. However, in some locations, the initial damage is compounded by delayed mortality, which can hinder mangrove recovery for extended periods following the storm. An illustrative case is the aftermath of Hurricane Irma, which incited one of the most substantial recorded instances of mangrove dieback in the region. Within the initial 15 months following Irma, a staggering 10,760 hectares of mangroves displayed signs of complete dieback, characterized by minimal or no regrowth [29–31].

Impoundments constructed during the period of 1940 to 1960 disrupted tidal circulation, leading to prolonged flooding, and resulting in a 76% decline in mangrove forests within the Indian River Lagoon (IRL) [25]. In response, rotational impoundment management (RIM) techniques were developed and introduced during the 1980s. These techniques enabled the natural cyclical flow of tidal waters into the impounded mangrove forests, on a rotational basis. This management intervention helped to recover the mangrove forests in this region. Similarly, during the 1950s and 1960s, predominantly along the central east coast of Florida, the common practice involved dredging estuarine sediments and depositing them onto coastal wetlands. This practice led to significant modifications in mangrove distribution. In some extensively ditched regions, up to 80% of mangrove forests were substituted with ditches and spoil piles. To counteract these impacts, the rehabilitation of coastal wetlands affected by dragline ditches was initiated in 1999. Since then, mangrove recovery has been observed in these areas.

Within the Ten Thousand Islands region, a notable occurrence of mangrove migration towards inland areas was observed. Thorough examination conducted by this study, along with research conducted by Krauss et al. (2011) [32], underscored that this transformation results from a combination of factors. These include the interplay of relative sea level rise, the establishment of waterways, localized precipitation patterns, and decreased freshwater discharge. The creation of waterways played a pivotal role in facilitating the dispersion of propagules necessary for the process of natural regeneration.

Likewise, a multifaceted and divergent pattern of mangrove alteration was evident in three neighboring sites within the Everglades [15]. Through their analysis of aerial photographs and images spanning from 1928 to 2004, Smith et al. (2013) [15] determined that fire contributed to the expansion of mangroves into salt marshes in one of the sites. However, in another site affected by fire, no discernible change was observed. In the third site, mangrove expansion was noted even in the absence of any notable fire incidents [15].

Significant transformations took place in Louisiana, particularly within the northernmost boundary spanning from 29° to 30.5° latitude. Giri et al. [17] demonstrated that a severe freeze occurring in late December of 1983 which led to a drastic reduction in Louisiana's mangrove coverage by approximately 90%. Subsequent harm from repeated sub-freezing temperatures in 1985, 1989, and 2010 impeded the recovery of the mangrove extent until the late 1990s and the following decades. Despite experiencing less frequent winter freeze events in the past ten years, Louisiana's mangrove coverage has not regained its historical extent from 1983. This is likely attributed to accelerated coastal retreat coupled with coastal subsidence, along with rapid sea level rise at a rate of 9.03 mm/year. Notably, Louisiana boasts the highest rates of relative sea level rise within the Gulf of Mexico.

Figure 10 illustrates the impact of a winter freeze on mangrove mortality and the subsequent hindered recovery caused by subsidence. By 2010, nearly all of Timbalier Island in southern Louisiana had become submerged, potentially attributed to rising sea levels.

In more recent times, there has been a noticeable emergence of red mangroves in previously unoccupied regions within Texas. Over the years, the expansion of mangrove forests in Texas has been propelled by relatively mild winters and potentially by rising sea levels. Within the timeframe of 1980 to 2015, the mangrove-covered area escalated by 2259 hectares, reflecting a remarkable 234% increase. An examination of changes indicated that the growth of mangroves predominantly occurred at the cost of salt marshes. A significant mangrove expansion was particularly notable between latitudes 27°30′ and 28°30′. In these specific zones, the expansion of mangroves primarily encroached upon existing marsh or upland areas. Within the last 15 years, new stands of mangroves emerged in the Corpus Christi Pass region.

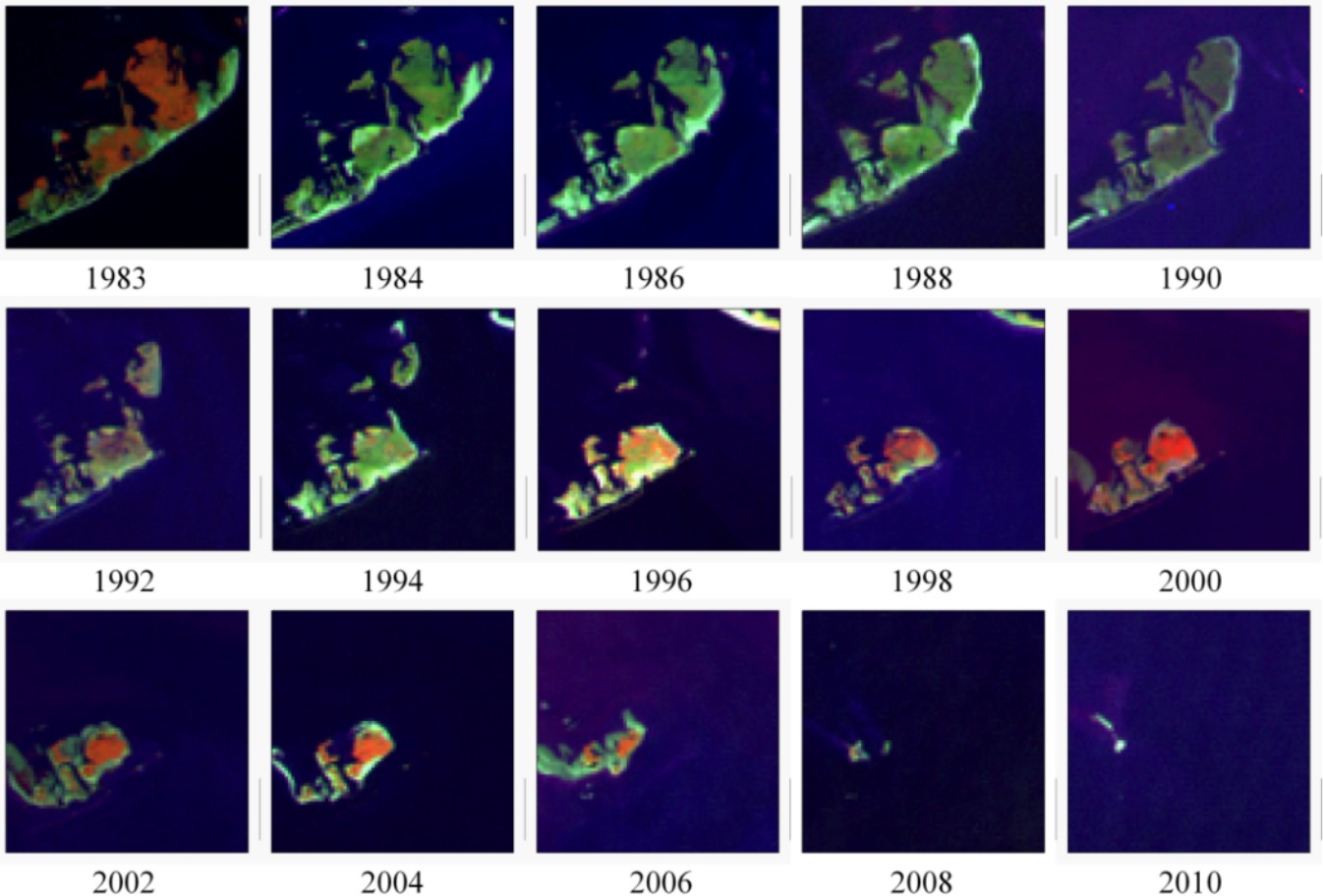

**Figure 10.** Extensive damage of mangrove forest due to the winter freeze of 1983 and subsequent recovery and near disappearance of the Timbalier Island, Louisiana.

### 3.4. Results Validation

While validating mangrove cover products using rigorous sampling methods and high-quality concurrent reference data is unquestionably preferable, resource constraints frequently hinder the attainment of fully comprehensive quantitative validation [33]. Nevertheless, we undertook a three-step validation process for our results: (i) visual comparison of our outcomes with existing mangrove maps, (ii) qualitative assessment, and (iii) quantitative validation.

In the initial method, we assessed the reliability of our database by comparing it with other available mangrove datasets covering both the CONUS and local areas. For the second approach, spanning all years, we partitioned the entire area into systematically distributed grids. Subsequently, we conducted visual inspections of each grid to detect and rectify any noticeable errors within the classified maps. Gross errors identified were corrected. These two processes not only contributed to a qualitative evaluation of the map but also enhanced the overall classification quality, while preserving consistency across the years.

During the assessment for the year 2020, we attained an overall accuracy of 95%. The producer accuracy and user accuracy for mangrove were 100% and 92%, respectively. The error matrix is provided in Table 3 below.

**Table 3.** Error matrix for 2020 classification.

| Classified Data | Reference Data | | | Producers Accuracy | Users Accuracy |
|---|---|---|---|---|---|
| | Mangrove | Non-mangrove | Water | | |
| Mangrove | 23 | 1 | 1 | 100.00 | 92.00 |
| Non-mangrove | 0 | 35 | 1 | 92.11 | 97.22 |
| Water | 1 | 2 | 37 | 95.87 | 94.87 |
| | | | | Overall Accuracy = 95% | |

## 4. Limitations

A significant drawback of the current study is the absence of high-quality Landsat satellite data for the intended target year. This limitation is especially pertinent in the case of earlier years, notably 1980 and 1985. This is because Landsat series, such as Landsat 4 and 5, lacked comprehensive global coverage for the desired target date. Consequently, the creation of a mosaic devoid of cloud cover necessitated the integration of data spanning multiple years. Additionally, a mechanical malfunction in the Scan-Line Corrector (SLC-Off) on the Landsat-7 satellite, causing a data loss ranging from 22% to 25% with each captured image, further compounded the difficulties in preparing a mosaic for the target date.

The availability of consistent, error-free, and cohesive composite images becomes vital for the consistent assessment and monitoring of mangrove forests over time. The five-year composites spanning from 1980 to 2020 demonstrate a remarkable seamlessness, unaffected by factors like atmospheric interference and sensor anomalies, such as the SLC-Off issue with Landsat-7, as well as gaps in pixel data. However, some of the changes were not captured on those mosaics.

Landsat satellite data with a spatial resolution of 30 meters did not facilitate the mapping of smaller mangrove patches. Mapping these smaller patches might require the utilization of very high-resolution satellite data, with spatial resolutions ranging from 1 to 5 m. It is important to note that larger and more uniform mangrove areas were mapped with greater accuracy compared to smaller, more complex, and diverse landscapes.

During the change analysis, we conducted a comparison of two classified thematic maps, which introduced errors into the change analysis process. This outcome reflects the fact that land-use/land-cover maps derived from remotely-sensed data inherently contain errors. The presence of cloud-free satellite data, pre-processing errors, and classification errors all contribute to limitations in post-classification change analysis.

## 5. Conclusions

We have compiled a comprehensive database detailing the distribution of mangroves across the entire CONUS at five-year intervals spanning from 1980 to 2020. This database was created utilizing Landsat satellite data with a spatial resolution of 30 m. Our findings unveiled that in the year 2020, the total extent of mangrove forests in the CONUS encompassed 266,179 hectares, with Florida accounting for 98.0% of the mangrove area, Louisiana contributing 0.6%, and Texas comprising 1.4%. The analysis of latitudinal distribution revealed a consistent trend of decreasing mangrove occurrence with the progression from south to north. Furthermore, our results indicated that approximately 85% of the total mangrove area within the CONUS was concentrated within the latitude range of 24.5° to 26.0° [17].

After conducting post-classification change analysis, it became evident that the overall mangrove area across the CONUS has experienced a 13.5% expansion from 1980 to 2020. Additionally, our findings indicate that the northernmost latitudinal boundaries of mangrove forests along the eastern and western coastlines of Florida, as well as in Louisiana and Texas, have not exhibited a systematic poleward expansion. While our analysis spanning

40 years has offered valuable insights into the contemporary drivers behind the range fluctuations, it is essential to acknowledge that this timeframe is relatively short. As a result, the historical dynamics of mangroves before the satellite era remain unknown.

The most notable increases in mangrove area from 1980 to the present were observed in the southern regions of Florida and Texas, well within their respective northern boundaries. The underlying causes of these changes were found to be location-specific rather than universal. A multitude of factors, including but not limited to sea level rise, the presence or absence of sub-freezing temperatures, alterations in land use, impoundment and dredging activities, shifts in hydrology, occurrences of fire, storms, sedimentation, erosion, and deliberate mangrove planting, collectively contribute to the observed changes. Particularly, the relatively milder winters and the reduced occurrence of sub-freezing temperatures in recent decades appear to facilitate local expansion.

Our findings underscore the intricate interplay of various forces influencing the northern limits of mangrove distribution. They underscore the necessity for sustained, long-term monitoring of this system's dynamics, especially as its significance grows in the context of adapting to rising sea levels and mitigating the impacts of heightened atmospheric $CO_2$. The compiled mangrove database could serve as a valuable tool for monitoring the potential impacts of climate change on mangrove forests in the future.

**Author Contributions:** Conceptualization C.G., methodology C.G. and J.L., software, validation and formal analysis J.L. and P.P., writing-original draft preparation and review and editing C.G. and P.P. All authors have read and agreed to the published version of the manuscript.

**Funding:** This research was funded by the U.S. Environmental Protection Agency through its Office of Research and Development (ORD) as a contribution to ORD's Safe and Sustainable Water Resources National Research Program. The APC was funded by the ORD.

**Data Availability Statement:** All the datasets generated from this study are freely available from www.epa.gov (Accessed on 13 June 2023).

**Acknowledgments:** We would like to extend our sincere thanks to Jeremy Schroeder who helped to analyze part of the 2020 Texas classification.

**Conflicts of Interest:** The author declares no conflict of interest.

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
