# Peer review of "Mangrove Forest Cover Change in the Conterminous United States from 1980–2020"

_remotesensing, doi:10.3390/rs15205018_

Round 1
Reviewer 1 Report
Please see the attached comments.

Basic english editing may be required. This has not been fully verified, given the extent of the scientific errors.
Author Response
I appreciate that the reviewer took time to comments particularly on accuracy assessment. They were very useful. Here are my responses:
- This ‘error matrix’ is not properly aligned and does not show corresponding producers and users accuracies for the classes ‘mangrove’, ‘non-mangrove’ and ‘water’. It is now corrected and properly aligned. My apology for the typo.
- The users and producers accuracies are not correctly quantified. No class has achieved a 100% accuracy. With the proper alignment, this issue is now resolved.
• An overall accuracy is not quantified. Overall accuracy is included in the error matrix
• The number of validation samples per class, is not valid. Such an area (extent of the study region) dictates far more reference samples. The creation and distribution of these reference samples must also be more thoroughly discussed. The discussion is added in line 177 methods section.
• This accuracy assessment, for the 2020 year says nothing about the accuracy of the other maps (quantifications) or the accuracy of the change over time, which should be independently evaluated. On line 430, you write: “It is presumed that comparable accuracy was attained for all other years from 1980-2015. Error matrix is provided below.” This is an invalid and impossible assumption. This statement is not removed.
Reviewer 2 Report
The paper addresses the important development of mangroves in the CONUS area. It analyses long time series from Landsat satellites using standard methods provided by the Google Earth Engine. The methodology is well described and the validation of results is appropriate.
Results are well presented and discussed addressing the impact of weather (i.e. winter freeze) and other natural and man-made effects.
The paper is well written with some minor issues to be corrected., which are listed in the following:
Line 23: The abbreviation "CONUS" is used the first time without explanation. This comes later in the text (i.e. at line 61) but then it alterates in the following paragraphs between the full expression and the abbreviation and even both several times. It is recommended to use the abbreviation from line 61 only.
Line 123: "ROI" - First the abbreviation is not explained and then it is unclear how and based on what information the ROIs were defined.
Table 1: at the beginning of lines 1 and 2 characters "S" and "N" appear without meaning. Please delete or explain.
Table 3: The error matrix needs to be corrected: (1) values for Line "Total" are missing and (2) in column "Classified Data" classes need to be moved 1 line down in order to correspond correctly between "Classified" and "Reference" data.
There are only minor spelling errors.
Author Response
Thank you very much for your comments. Here are my responses:
Line 23: The abbreviation "CONUS" is used the first time without explanation. This comes later in the text (i.e. at line 61) but then it alterates in the following paragraphs between the full expression and the abbreviation and even both several times. It is recommended to use the abbreviation from line 61 only.
Agreed and corrected
Line 123: "ROI" - First the abbreviation is not explained and then it is unclear how and based on what information the ROIs were defined.
ROI is defined in Line 18
It is explained in line 107 and 108
Table 1: at the beginning of lines 1 and 2 characters "S" and "N" appear without meaning. Please delete or explain.
deleted
Table 3: The error matrix needs to be corrected: (1) values for Line "Total" are missing and (2) in column "Classified Data" classes need to be moved 1 line down in order to correspond correctly between "Classified" and "Reference" data.
corrected
thank you
Reviewer 3 Report
The article falls into the scope of Remote Sensing Journal, and it focuses on a spatiotemporal analysis of Landsat satellite images for assessing changes in mangrove forest in United States.
Abstract is well written with main results mentioned in this section.
Introduction should contain more references focusing not only on mangrove forest issues, but also the remote sensing and satellite data analysis.
Study Area, Data Basis, and Methods chapter is clearly written but the text needs minor formal corrections (e.g. ERDAS Imagine vs. imagine, withthe goal - and others…), not only in this chapter.
Results and Discussion – L207 – the abbreviation CONUS has been already explained in previous chapter.
Figure 3 – use the same unites in all maps – km vs miles
Results are well written.
Conclusion is in line with the aims defined in Introduction.
Maybe I explained well why I recommend to publish it with major revision.
Author Response
Introduction should contain more references focusing not only on mangrove forest issues, but also the remote sensing and satellite data analysis.
Reference on the remote sensing and satellite data analysis are included
Study Area, Data Basis, and Methods chapter is clearly written but the text needs minor formal corrections (e.g. ERDAS Imagine vs. imagine, withthe goal - and others…), not only in this chapter.
It is now consistent ERDAS Imagine
Results and Discussion – L207 – the abbreviation CONUS has been already explained in previous chapter.
Corrected
Figure 3 – use the same unites in all maps – km vs miles
Excellent point, the grid map is for reference only
thank you very much.
Reviewer 4 Report
Dear Authors,
Greetings,
I find the topic you've chosen to be quite intriguing and valuable. I have included my general comments in the attached file for your review.
Best regards,

Author Response
Thank you for the reviewer for the comments and suggestions.
Comments and suggestions marked were incorporated.
thanks
Round 2
Reviewer 1 Report
The major flaws in the accuracy assessment have not been fixed. An area of over 250,000 ha cannot be validated using only 25 samples per class. There is also little to no confidence that the given validation points were correctly created.
Additionally, given a valid assessment of the accuracy of the current map was conducted, this says nothing to the change over time.
Will be evaluated after scientific review is satisfied.
Reviewer 3 Report
From my point of view, the authors corrected the contribution acording to reviewers recommendations. I am satisfied.